# Types and clinical outcomes of chemical ingestion in emergency departments in South Korea (2011-2016)

**Jae Hee Lee, Duk Hee Lee**  *

Department of Emergency Medicine, Ewha Womans University, Seoul, South Korea

* ewhain78@gmail.com

## Abstract

### Objective

This study aims to provide basic data on the types and frequency of chemical ingestions and the clinical outcomes of chemical ingestion injury.

### Methods

This study retrospectively analyzed the data obtained from the Emergency Department-Based Injury In-depth Surveillance of the Korea Centers for Disease Control and Prevention (South Korea) from 2011 to 2016. Patients ingesting chemicals aged $\geq$ 18 years were included, but those ingesting unknown chemical substances or with unknown clinical outcomes were excluded.

### Results

This study included 2,712 (47.2% were men and 52.8% were women, mean age, 47.05 years) patients ingesting chemicals. Unintentional and intentional ingestions were reported in 1,673 (61.7%) and 1,039 (38.3%), respectively. The most commonly ingested chemical substances were hypochlorites, detergents, ethanol, and acetic acid. In the unintentional ingestion group, the most common chemicals upon admission were hypochlorites (74), glacial acetic acid (60), and detergent (33). The admission rates were 60% for glacial acetic acid, 58.3% ethylene glycol, and 30.4% other alkali agents. In the intentional ingestion group, the most common chemicals upon admission were hypochlorites (242), glacial acetic acid (79), ethylene glycol (42), and detergent (41). The admission rates were 91.9% for glacial acetic acid, 87.5% ethylene glycol, 85.7% potassium cyanide, and 81.4% hydrochloric acid. In total, 79 deaths (10 unintentional ingestions, 69 intentional ingestion) were reported, and glacial acetic acid had an odds ratio of 9.299 for mortality.

### Conclusion

We compared the intentional and unintentional ingestion groups, and analyzed the factors affecting hospital admission and mortality in each group. The types and clinical outcomes of

**Data Availability Statement:** The Korean Center for Disease Control (KCDC) is the authority for accessing the data analyzed, and there are ethical restrictions on sharing a dataset because the data contain potentially identifying information. The

KCDC (http://www.cdc.go.kr) can be contacted for data access via the Injury research team email (kcdcinjury@korea.kr) or by calling 82-43-719-7407. The authors used the dataset "Emergency Department-Based Injury In-depth Surveillance of the Korea Centers for Disease Control and Prevention (KCDC)." The authors did not have special access privileges.

**Funding:** This work was supported by the National Research Foundation of Korea (NRF) grant funded by the Korea government (MSIT) (No. 2018R1C1B5046096), and a fund by Research of Korea Centers for Disease Control and Prevention (Emergency Department-Based Injury In-depth Surveillance). The funders had no role in study design, data collection and analysis, decision to publish, or preparation of the manuscript.

**Competing interests:** The authors have declared that no competing interests exist.

chemical ingestion varied depending on the purpose of chemical ingestion. The findings are considered beneficial in establishing treatment policies for patients ingesting chemicals.

## Introduction

With the advancement of manufacturing and industry, approximately 120,000 chemicals are being distributed worldwide, with approximately 40,000 of them being distributed in South Korea [1]]. Moreover, as the distribution of chemicals increases, exposure to these chemicals also increases. In the United States, an estimated 10,000 chemical exposures are reported annually. occur each year [[2,3]. During chemical exposure, both the exposure and the victims should be properly monitored. Exposure events are primarily monitored by the departments responsible for managing chemicals, and data collection from the victims is hospital- or poison center-based [4–7]]. In South Korea, data regarding chemical exposures are limited to the exposure events, such as chemical accidents or chemical exposures at the workplace [1,8–10] In this study, we aim to provide basic data on chemical ingestion, including the types of chemicals and frequency of ingestion and clinical outcomes of chemical ingestion. Therefore, we compared the characteristics of chemical ingestion according to their intentionality and analyzed the factors that affect the poor prognosis in each group.

## Materials and methods

### Setting and data collection

This retrospective observational study was conducted with the approval of the institutional review board of Ewha Womans' University Mok-dong hospital (IRB No.2019-08-006). The informed consent for gathering the data related to the emergency department (ED) visit was obtained from all participating patients or patients' guardians by the Korea Centers for Disease Control and Prevention (KCDC) according to national research committee. This study analyzed the data from the Emergency Department-Based Injury In-depth Surveillance under the supervision of the KCDC from 2011 to 2016. The Emergency Department-Based In-depth Surveillance is a data collection system that proactively collects data on basic epidemiology, treatment, and outcomes of injured patients presenting to the ED. Implemented in 2006, the collection system involved 20 agencies from 2011 to 2014 and 23 agencies since 2015. Each hospital employs personnel who are responsible for data entry and quality control and who are regularly trained and supervised by the KCDC.

In this study, the definition of chemical ingestion was as follows: consumption of an artificial toxic substance (according to the toxic substance classification) that caused poisoning and injury in patients who visited the ED. All study subjects were aged older than 18 years. Excluded cases were subjects aged younger than 17 years and subjects with missing information on intentionality or ED clinical outcomes.

### Outcome measures

According to the injury surveillance system, in cases of poisoning, the trained personnel must classify the substance according to code and enter the proper name (general name or product name). In this study, two emergency physicians and one chemist identified and categorized the main components of the ingested chemicals. All patients ingesting chemicals were classified into either the intentional ingestion group or the unintentional ingestion group, and

comparison and subgroup analysis were performed between the two groups. Sex, age, mode of ED arrival, and insurance type were compared as the general features. The analyzed characteristics related to injury were as follows: period from the time of injury to ED presentation, place where the injury occurred, activity at the time of injury, alcohol ingestion before injury, date of the visit, time of the visit, and duration of ED stay. The results of ED treatment (discharge from ED, admission to general ward (GW) or intensive care unit [ICU]) and mortality were used to compare the severity of injuries.

## Statistical analysis

We compared and analyzed the general and injury-related characteristics between the intentional and unintentional ingestion groups. The mean with standard deviation and median with 25th and 75th percentiles were expressed as the continuous variables. The number of patients and percentage were expressed as the noncontinuous variables. For items requiring statistical verification, an independent T-test and Mann-Whitney U test were used for the continuous variables. A chi-squared test was used for the noncontinuous variables. Using cross-analysis and average comparison, we selected the variables that showed significant differences between the two groups, and based on the researchers' discretion, we selected the variables considered as factors for hospital admission and mortality. Multiple logistic regression using backward stepwise selection (likelihood ratio) was performed, and the remaining variables with p-value less than 0.05 were listed in the table. The statistics program Statistical Package for the Social Sciences Statistics for Windows version 21 (International Business Machines Corporation, Armonk, NY, USA) was used, and 95% confidence intervals were considered statistically significant.

## Results

From 2011 to 2016, a total of 4,741 cases of chemical ingestion were reported. After applying the exclusion criteria, this study included 2,712 patients ingesting chemicals (1,247 patients aged younger than 18 years, 644 patients with missing data for ED treatment results, 138 patients with missing data for intentionality). A total of 52.8% of the patients were women. The mean age was 47.05 years. The unintentional ingestion and intentional ingestion groups were composed of 1,673 (61.7%) and 1,039 patients (38.3%), respectively.

The comparison results of the unintentional and intentional ingestion groups are presented in Table 1. The proportion of women in the intentional ingestion group (66.3%) was higher than that in the unintentional ingestion group (44.5%). Chemical ingestion was reported in homes in 88.8% of patients in the intentional ingestion group and in 53% of patients in the unintentional ingestion group (p<0.0001). The activity at the time of injury was described as "other(unspecified)" in 95.7% of patients in the intentional ingestion group, while daily living activity was noted in 48.2% of patients in the unintentional ingestion group. Alcohol was ingested before the injury in 43.1% of patients in the intentional ingestion group, a percentage significantly higher than the 12.6% in the unintentional ingestion group. ED treatment results showed that 79.3% of patients in the unintentional ingestion group were discharged from the ED. Patients in the intentional ingestion group had significantly larger proportions of GW admission, ICU care, and mortality in the ED than the unintentional ingestion group. The total number of deaths from chemical ingestion was 79, comprising 10 from the unintentional ingestion group and 69 from the intentional ingestion group.

The subgroup analysis of patients ingesting chemicals was performed according to intentionality. Regarding the agents ingested in 1,673 patients in the unintentional ingestion group, hypochlorite-related agents were most common (410), followed by detergents, ethanol,

**Table 1. General characteristics of chemical ingestion between unintentional and intentional ingestion groups.**

| | Unintentional | | Intentional | | Total | | P value |
|---|---|---|---|---|---|---|---|
| | N | % | N | % | N | % | |
| No. of patients | 1673 | 61.7 | 1039 | 38.3 | 2712 | 100 | |
| Sex | | | | | | | |
| Male | 929 | 55.5 | 650 | 33.7 | 1279 | 47.2 | <0.0001 |
| Female | 744 | 44.5 | 689 | 66.3 | 1433 | 52.8 | |
| Age (yrs, mean±SD) | 47.33±16.91 | | 46.51±17.48 | | 47.05±17.24 | | 0.226 |
| Age distribution in 10 years | | | | | | | <0.0001 |
| 20–30 | 332 | 19.8 | 210 | 20.2 | 542 | 20.0 | |
| 31–40 | 315 | 18.8 | 234 | 22.5 | 549 | 20.2 | |
| 41–50 | 314 | 18.8 | 202 | 19.4 | 516 | 19.0 | |
| 51–60 | 350 | 20.9 | 185 | 17.8 | 535 | 19.7 | |
| 61–70 | 179 | 10.7 | 64 | 6.2 | 243 | 9.0 | |
| 71–80 | 121 | 7.2 | 99 | 9.5 | 220 | 8.1 | |
| ≥81 | 62 | 3.7 | 45 | 4.3 | 107 | 3.9 | |
| Mode of arrival | | | | | | | 0.420 |
| Walk-in (include car, foot, etc.) | 1266 | 75.7 | 817 | 78.6 | 2083 | 76.8 | |
| 119 | 238 | 14.2 | 132 | 12.7 | 370 | 13.6 | |
| Private ambulance | 162 | 9.7 | 86 | 8.3 | 248 | 9.1 | |
| Police | 0 | .0 | 1 | .1 | 1 | .0 | |
| Air | 4 | .2 | 1 | .1 | 5 | .2 | |
| Others | 1 | .1 | 1 | .1 | 2 | .1 | |
| Unknown | 2 | .1 | 1 | .1 | 3 | .1 | |
| Time interval from injury to ED visit (hrs, median, 25%, 75%) | 1 (0, 5) | | 1 (0, 4) | | 1 (0, 5) | | 0.002[a] |
| Insurance | | | | | | | 0.812 |
| National health insurance | 1410 | 84.3 | 887 | 85.4 | 2297 | 84.7 | |
| Vehicle | 190 | 11.4 | 112 | 10.8 | 302 | 11.1 | |
| Medicaid beneficiary | 30 | 1.8 | 20 | 1.9 | 50 | 1.8 | |
| Self-pay (uninsured) | 35 | 2.1 | 17 | 1.6 | 52 | 1.9 | |
| Others | 8 | .5 | 3 | .3 | 11 | .4 | |
| Where the poisoning occurred | | | | | | | <0.0001 |
| House | 887 | 53.0 | 914 | 88.0 | 1801 | 66.4 | |
| Residential facility | 19 | 1.1 | 17 | 1.6 | 36 | 1.3 | |
| Medical facility | 28 | 1.7 | 21 | 2.0 | 49 | 1.8 | |
| School, education facility | 57 | 3.4 | 3 | .3 | 60 | 2.2 | |
| Sport facility | 9 | .5 | 1 | .1 | 10 | .4 | |
| Road | 36 | 2.2 | 9 | .9 | 45 | 1.7 | |
| Transportation area except road | 3 | .2 | 6 | .6 | 9 | .3 | |
| Factory, industrial facility | 291 | 17.4 | 4 | .4 | 295 | 10.9 | |
| Farm | 32 | 1.9 | 2 | .2 | 34 | 1.3 | |
| Amusement, cultural public facility | 26 | 1.6 | 7 | .7 | 33 | 1.2 | |
| Commercial facility | 238 | 14.2 | 42 | 4.0 | 280 | 10.3 | |
| Outdoor, river, sea | 24 | 1.4 | 9 | .9 | 33 | 1.2 | |
| Other | 1 | .1 | 1 | .1 | 2 | .1 | |
| Unknown | 22 | 1.3 | 3 | .3 | 25 | .9 | |
| Activity | | | | | | | <0.0001 |
| Work | 434 | 25.9 | 0 | .0 | 434 | 16.0 | |
| Unpaid labor | 214 | 12.8 | 3 | .3 | 217 | 8.0 | |

*(Continued)*

**Table 1.** (Continued)

|  | Unintentional | | Intentional | | Total | | P value |
|---|---|---|---|---|---|---|---|
|  | N | % | N | % | N | % |  |
| Education | 34 | 2.0 | 0 | .0 | 34 | 1.3 | |
| Exercise | 6 | .4 | 0 | .0 | 6 | .2 | |
| Leisure | 123 | 7.4 | 1 | .1 | 124 | 4.6 | |
| Daily living activity | 806 | 48.2 | 41 | 3.9 | 847 | 31.2 | |
| Hospital treatment | 7 | .4 | 0 | .0 | 7 | .3 | |
| Travel | 1 | .1 | 0 | .0 | 1 | .0 | |
| Other | 33 | 2.0 | 994 | 95.7 | 1027 | 37.9 | |
| Unknown | 15 | .9 | 0 | .0 | 15 | .6 | |
| Alcohol ingestion before injury | | | | | | | <0.0001 |
| No | 1313 | 78.5 | 487 | 46.9 | 1800 | 66.4 | |
| Yes | 210 | 12.6 | 448 | 43.1 | 658 | 24.3 | |
| Unknown | 150 | 9.0 | 104 | 10.0 | 254 | 9.4 | |
| Day of presentation | | | | | | | 0.095 |
| Weekday (Mon-Thu) | 913 | 54.6 | 601 | 57.8 | 1514 | 55.8 | |
| Weekend (Fri-Sun) | 760 | 45.4 | 438 | 42.2 | 1198 | 44.2 | |
| Time of presentation | | | | | | | <0.0001 |
| Day | 527 | 31.5 | 280 | 26.9 | 807 | 29.8 | |
| Evening | 810 | 48.4 | 390 | 37.5 | 1200 | 44.2 | |
| Night | 336 | 20.1 | 369 | 35.5 | 705 | 26.0 | |
| ED stay (hrs, mean±SD) | 40.50 ±108.17 | | 24.21±63.34 | | 33.79±92.91 | | <0.0001 |
| ED treatment result | | | | | | | <0.0001 |
| Discharge | 1327 | 79.3 | 482 | 46.4 | 1809 | 66.7 | |
| General ward | 261 | 15.6 | 328 | 31.6 | 589 | 21.7 | |
| Intensive care unit | 83 | 5.0 | 194 | 18.7 | 277 | 10.2 | |
| Death | 2 | .1 | 35 | 3.4 | 37 | 1.4 | |
| Hospital mortality[b] | 10 | 0.60 | 69 | 6.60 | 79 | 2.90 | <0.0001 |

[a]Mann-Whitney test

[b]Hospital mortality included ED death.

toluene, sodium hydroxide, and glacial acetic acid. The most common chemicals upon admission in the unintentional ingestion group were hypochlorites in 74 patients, glacial acetic acid in 60, and detergent in 33. The admission rates were as follows: 60% for glacial acetic acid, 58.3% ethylene glycol, and 30.4% other alkali agents. Ten deaths were noted in the unintentional ingestion group due to glacial acetic acid (3), other hydrocarbons (2), hypochlorite-related agent (1), and detergent (1), toluene (1), methanol (1), and hydrochloric acid (1) (Table 2). The different characteristics between the admitted and discharged patients in the unintentional ingestion group were compared and are listed in Table 3. The proportion of men in the admission group was significantly higher than that in the discharged group, and 23.04% of men and 17.74% of women were admitted. The mean age of the admission group was higher (54.01 years) than that of the discharge group (45.59 years). Regression analysis of risk factors for admission in the unintentional ingestion group revealed that male sex and increased age were significant factors. Among the ingested chemicals, ethylene glycol, glacial acetic acid, sulfuric acid, methanol, and other acidic substances significantly increased the risk for admission. Only consumption of hydrofluoric acid was associated with decreased risk of admission (Table 4).

**Table 2. Common agents of unintentional and intentional chemical ingestion in adult at ED, South Korea, 2011–2016.**

| Unintentional | | | | Intentional | | | |
|---|---|---|---|---|---|---|---|
| Chemical | Total | Admission | Death | Chemical | Total | Admission | Death |
| | (n = 1673) | (n = 346) | (n = 10) | | (n = 1039) | (n = 557) | (n = 69) |
| Hypochlorites related agents | 410 | 74 | 1 | Hypochlorites related agents | 514 | 242 | 9 |
| | | 18.0% | 0.2% | | | 47.1% | 1.8% |
| Detergents (soaps-anionic and nonionic) | 209 | 33 | 1 | Detergents (soaps-anionic and nonionic) | 121 | 41 | 6 |
| | | 15.8% | 0.5% | | | 33.9% | 5.0% |
| Ethanol | 154 | 21 | (-) | Glacial acetic acid | 86 | 79 | 33 |
| | | 13.6% | 0.0% | | | 91.9% | 38.4% |
| Toluene | 119 | 19 | 1 | Ethylene glycol | 48 | 42 | 1 |
| | | 16.0% | 0.8% | | | 87.5% | 2.1% |
| Sodium hydroxide | 100 | 23 | (-) | Hydrochloric acid | 43 | 35 | 6 |
| | | 23.0% | 0.0% | | | 81.4% | 14.0% |
| Glacial acetic acid | 100 | 60 | 3 | Ethanol | 41 | 10 | 1 |
| | | 60.0% | 3.0% | | | 24.4% | 2.4% |
| Other hydrocarbon | 76 | 18 | 2 | Sodium hydroxide | 39 | 26 | 3 |
| | | 23.7% | 2.6% | | | 66.7% | 7.7% |
| Hydrofluoric acid | 66 | 3 | (-) | Potassium Cyanide | 21 | 18 | 3 |
| | | 4.5% | 0.0% | | | 85.7% | 14.3% |
| Hydrogen peroxide | 37 | 4 | (-) | Methanol | 15 | 10 | (-) |
| | | 10.8% | 0.0% | | | 66.7% | .0% |
| Sulfuric acid | 35 | 2 | (-) | Toluene | 15 | 9 | 1 |
| | | 5.7% | 0.0% | | | 60.0% | 6.7% |
| Methanol | 29 | 11 | 1 | Sodium percarbonate | 13 | 3 | (-) |
| | | 37.9% | 3.4% | | | 23.1% | 0.0% |
| Hydrochloric acid | 28 | 6 | 1 | Other hydrocarbon | 10 | 6 | 2 |
| | | 21.4% | 3.6% | | | 60.0% | 20.0% |
| Ethylene glycol | 24 | 14 | (-) | Acetone | 9 | 1 | (-) |
| | | 58.3% | 0.0% | | | 11.1% | 0.0% |
| Other alkali agent | 23 | 7 | (-) | Hydrogen peroxide | 8 | 3 | (-) |
| | | 30.4% | 0.0% | | | 37.5% | 0.0% |

The most commonly ingested chemicals of the 1,039 patients in the intentional group were hypochlorite-related agents (514), followed by detergent, glacial acetic acid, ethylene glycol, and hydrochloric acid. A total of 242 patients were admitted following the ingestion of hypochlorites, 79 for glacial acetic acid, 42 for ethylene glycol, and 41 for detergent. The admission rates were as follows: 91.9% for glacial acetic acid, 87.5% ethylene glycol, 85.7% potassium cyanide, and 81.4% hydrochloric acid. Among the 69 deaths in the intentional ingestion group, 33 patients ingested glacial acetic acid, 9 ingested hypochlorites, 6 detergents, and 6 hydrochloric acids (Table 2). The different characteristics of patients admitted and those discharged in the intentional chemical ingestion group were compared (Table 3). The mean age of patients in the admission group was higher (49.94 years) than that of the discharge group (42.56 years). Regression analysis of risk factors for admission in the intentional ingestion group demonstrated significantly increased risk of admission with older age and consumption of sodium hydroxide, ethylene glycol, glacial acetic acid, hydrochloric acid, hypochlorites, methanol, or potassium cyanide (Table 4).

Sixty-nine patients in the intentional ingestion group died, and a comparison of characteristics between patients who survived and those who died revealed an older mean age in the

**Table 3. Comparison of general characteristics between discharge and admission groups in unintentional and intentional chemical ingestion.**

| | Unintentional | | | | | | | Intentional | | | | | | |
|---|---|---|---|---|---|---|---|---|---|---|---|---|---|---|
| | Discharge | | Admission | | Total | | p-value | Discharge | | Admission | | Total | | p-value |
| | N | % | N | % | N | % | | N | % | N | % | N | % | |
| No. of patients | 1327 | 79.3 | 346 | 20.7 | 1673 | 100 | | 482 | 46.4 | 557 | 53.6 | 1039 | 100 | |
| Sex | | | | | | | | | | | | | | |
| Male | 715 | 53.9 | 214 | 61.8 | 929 | 55.5 | 0.008 | 150 | 31.1 | 200 | 35.9 | 350 | 33.7 | 0.104 |
| Female | 612 | 46.1 | 132 | 38.2 | 744 | 44.5 | | 332 | 68.9 | 357 | 64.1 | 689 | 66.3 | |
| Age (yrs, mean±SD) | 45.59 ±16.14 | | 54.01 ±18.11 | | 47.33 ±16.91 | | <0.0001 | 42.56 ±15.16 | | 49.94 ±18.61 | | 46.51 ±17.48 | | <0.0001 |
| Time interval from injury to ED visit (hrs, median, 25%,75%) | 1 (0, 5) | | 2 (0, 7) | | 1 (0, 5) | | 0.008* | 1 (0, 4) | | 1 (0, 5) | | 1 (0, 4) | | <0.0001[a] |
| Alcohol ingestion before injury | | | | | | | 0.180 | | | | | | | <0.0001 |
| No | 1047 | 78.9 | 266 | 76.9 | 1313 | 78.5 | | 187 | 38.8 | 300 | 53.9 | 487 | 46.9 | |
| Yes | 157 | 11.8 | 53 | 15.3 | 210 | 12.6 | | 254 | 52.7 | 194 | 34.8 | 448 | 43.1 | |
| Unknown | 123 | 9.3 | 27 | 7.8 | 150 | 9.0 | | 41 | 8.5 | 63 | 11.3 | 104 | 10.0 | |
| Time of presentation | | | | | | | 0.005 | | | | | | | <0.0001 |
| Day | 393 | 29.6 | 134 | 38.7 | 527 | 31.5 | | 108 | 22.4 | 172 | 30.9 | 280 | 26.9 | |
| Evening | 661 | 49.8 | 149 | 43.1 | 810 | 48.4 | | 173 | 35.9 | 217 | 39.0 | 390 | 37.5 | |
| Night | 273 | 20.6 | 63 | 18.2 | 336 | 20.1 | | 201 | 41.7 | 168 | 30.2 | 369 | 35.5 | |
| ED stay (hrs) | 48.18 ±120.29 | | 11.02 ±0.77 | | 40.50 ±108.17 | | <0.0001 | 39.61 ±90.63 | | 10.89 ±0.86 | | 24.21 ±63.34 | | <0.0001 |

[a]Mann-Whitney test

**Table 4. Multivariate analysis of unintentional and intentional chemical ingestion for admission.**

| | Multivariate | | | | Univariate | | | |
|---|---|---|---|---|---|---|---|---|
| | OR | 95% C.I. for EXP(B) | | Sig. | OR | 95% C.I. for EXP(B) | | Sig |
| | | Lower | Upper | | | Lower | Upper | |
| Unintentional | | | | | | | | |
| Sex: male | 1.370 | 1.057 | 1.776 | 0.017 | 1.388 | 1.089 | 1.768 | 0.008 |
| Age | 1.025 | 1.017 | 1.032 | <0.0001 | 1.030 | 1.022 | 1.037 | <0.0001 |
| Ethylene glycol | 5.842 | 2.522 | 13.530 | <0.0001 | 5.554 | 2.445 | 12.614 | <0.0001 |
| Glacial acetic acid | 5.568 | 3.591 | 8.632 | <0.0001 | 6.750 | 4.435 | 10.274 | <0.0001 |
| Hydrofluoric acid | 0.281 | 0.087 | 0.908 | 0.034 | 0.175 | 0.055 | 0.562 | 0.003 |
| Sulfuric acid | 26.132 | 3.105 | 219.890 | 0.003 | 23.400 | 2.808 | 195.017 | 0.004 |
| Methanol | 3.219 | 1.464 | 7.077 | 0.004 | 2.388 | 1.117 | 5.104 | 0.025 |
| Other acid agent | 11.817 | 2.062 | 67.733 | 0.006 | 7.749 | 1.413 | 42.480 | 0.018 |
| Intentional | | | | | | | | |
| Age | 1.019 | 1.011 | 1.028 | <0.0001 | 1.026 | 1.018 | 1.033 | <0.0001 |
| Alcohol ingestion before injury: Yes | 0.469 | 0.355 | 0.621 | <0.0001 | 0.480 | 0.374 | 0.616 | <0.0001 |
| Sodium hydroxide | 3.159 | 1.515 | 6.590 | 0.002 | 1.766 | 0.897 | 3.477 | 0.100 |
| Ethylene glycol | 17.008 | 6.840 | 42.287 | <0.0001 | 6.470 | 2.726 | 15.357 | <0.0001 |
| Glacial acetic acid | 16.064 | 6.986 | 36.941 | <0.0001 | 11.215 | 5.124 | 24.545 | <0.0001 |
| Hydrochloric acid | 7.189 | 3.126 | 16.532 | <0.0001 | 3.973 | 1.825 | 8.650 | 0.001 |
| Hydrochlorites related agent | 1.623 | 1.170 | 2.252 | 0.004 | 0.593 | 0.464 | 0.759 | <0.0001 |
| Methanol | 3.626 | 1.180 | 11.148 | 0.025 | 1.744 | 0.592 | 5.138 | 0.313 |
| Potassium cyanide | 11.270 | 3.165 | 40.130 | <0.0001 | 5.332 | 1.561 | 18.214 | 0.008 |

**Table 5. General characteristics of intentional chemical ingestion between survival and mortality groups.**

| | Survival | | Mortality | | Total | | P value |
|---|---|---|---|---|---|---|---|
| | N | % | N | % | N | % | |
| No. of patients | 970 | 93.4 | 69 | 6.6 | 1039 | 100 | |
| Sex | | | | | | | |
| Male | 325 | 33.5 | 25 | 36.2 | 350 | 33.7 | 0.643 |
| Female | 645 | 66.5 | 44 | 63.8 | 689 | 66.3 | |
| Age (yrs, mean±SD) | 45.14±16.79 | | 65.86±15.49 | | 46.51±17.48 | | <0.0001 |
| Time interval from injury to ED visit (hrs, median, 25%, 75%) | 1 (0, 5) | | 1 (0, 2) | | 1 (0, 4) | | 0.724 |
| Alcohol ingestion before injury | | | | | | | <0.0001 |
| No | 443 | 45.7 | 44 | 63.8 | 487 | 46.9 | |
| Yes | 434 | 44.7 | 14 | 20.3 | 448 | 43.1 | |
| Unknown | 93 | 9.6% | 11 | 15.9 | 104 | 10.0 | |
| Time of presentation | | | | | | | <0.0001 |
| Day | 247 | 25.5 | 33 | 47.8 | 280 | 26.9 | |
| Evening | 366 | 37.7 | 24 | 34.8 | 390 | 37.5 | |
| Night | 357 | 36.8 | 12 | 17.4 | 369 | 35.5 | |
| ED stay (hrs, mean±SD) | 25.22±65.44 | | 10.00±0.00 | | 24.21±63.34 | | <0.0001 |

mortality group of 65.86 years compared with the 45.14 years of the survival group (Table 5). Regression analysis was performed to identify risk factors for mortality in the intentional ingestion group, age and glacial acetic acid ingestion significantly increased the risk of death. Alcohol and hypochlorite ingestion were shown to reduce the risk of death (Table 6). To determine the association between glacial acetic acid ingestion and age, the interaction was analyzed, and a significant result (p = 0.07) was observed. Regression analysis was performed by dividing patients into four groups according to age and glacial acetic acid ingestion. The risk of mortality increased to 25.477 times when glacial acetic acid was ingested by a patient younger than 65 years and 80.605 times when ingested by a patient 65 years or older (Supplement 2)

## Discussion

This study is the first study to analyze of data from patients ingesting chemicals who visited the ED. To identify the clinical features and risk factors of patients ingesting chemicals who visited the ED, we not only conducted a comparison between the intentional and unintentional ingestion groups but also analyzed the factors affecting hospital admission and mortality in each group. Each analysis was necessary because the characteristics of the intentional and unintentional ingestion groups were different. According to this study, hypochlorites and detergent were the most commonly ingested chemicals in both the intentional and unintentional ingestion groups. The next most common substances were ethanol, toluene, sodium hydroxide, and glacial acetic acid in the unintentional ingestion group and glacial acetic acid,

**Table 6. Multivariate analysis of intentional chemical ingestion for mortality.**

| | Multivariate | | | | Univariate | | | |
|---|---|---|---|---|---|---|---|---|
| | OR | 95% C.I. for EXP(B) | | Sig. | OR | 95% C.I. for EXP(B) | | Sig. |
| | | Lower | Upper | | | Lower | Upper | |
| Age | 1.055 | 1.036 | 1.073 | <0.0001 | 1.067 | 1.054 | 1.083 | <0.0001 |
| Glacial acetic acid | 9.299 | 4.446 | 19.448 | <0.0001 | 15.860 | 9.173 | 27.422 | <0.0001 |
| Hydrochlorites related agent | 0.401 | 0.168 | 0.960 | 0.040 | 0.138 | 0.068 | 0.281 | <0.0001 |

ethylene glycol, hydrochloric acid, and ethanol in the intentional ingestion group (Supplement). The 2017 American Association of Poison Control Centers (AAPCC) annual report showed that the most common poisoning categories in the United States, excluding therapeutic drugs, are cleaning substances, cosmetics/personal care products, foreign bodies/toys, plants, bites, chemicals, and fumes/gases/vapors, in that order. More than 10,000 individual substances were reported, including bleach, hypochlorites, other foreign bodies/toys, cream/lotion/makeup, desiccants, miscellaneous essential oils, glow products, toothpaste, deodorants, soaps, and laundry detergent/liquid [4].

In the present study, hypochlorite was the most commonly ingested chemical and had the highest number of hospital admissions. The United States reported a total of 40,302 cases of hypochlorite ingestion in 2017, the second highest frequency of exposure to a single substance [[4]]. Hypochlorite in combination with sodium or calcium is widely used for bleaching or discoloration and rust removal. For home use, the concentration is low (3%-5%), and it can irritate the skin without causing serious burns. On the contrary, the concentration of industrial hypochlorite is approximately 20%, which can cause severe corrosive induction and may also result in pulmonary irritation [[11]]. Hypochlorite was an admission risk factor in the intentional ingestion group in this study, but it was also associated with decreased risk of mortality considering that fatal side effects are influenced by the substance's concentration.

Among the admitted patients in this study, the second most commonly ingested substance was glacial acetic acid, widely used for industrial purposes. At concentrations of 10–25%, it acts as an irritant, but at concentrations of 25% or more, it can cause corrosive injury. Skin contact can cause burns or blisters, and mucosal exposure can cause tissue damage [12]]. Hence, glacial acetic acid in concentrations greater than 20% is classified and managed as a toxic substance in several countries. However, in South Korea, after the removal of heavy metals from industrial glacial acetic acid, it is allowed for use in food products as vinegar. Thus, glacial acetic acid in concentrations greater than 99% can be easily purchased at large supermarkets and grocery stores [13,14]]. Notably, glacial acetic acid was the fourth most commonly ingested substance in 2,712 patients ingesting chemicals in the present study. South Korea has a higher incidence of glacial acetic acid ingestion compared to other countries. For example, in the United States, a total of 1,068,976 cases of nontherapeutic drug exposures were reported for the whole of 2017, and only 7347 of these were due to acid exposure. Among them, the most commonly ingested acid was hydrochloric acid (1,839 cases) followed by hydrofluoric acid (676 cases) [[4]]. Considering that Koreans can easily access glacial acetic acid, accidental exposure is common, regardless if precautionary measures are provided by various media outlets. However, for individuals who are planning to intentionally ingest highly toxic materials, provision of this information is dangerous. Among patients ingesting chemicals who present to an ED in South Korea, glacial acetic acid ingestion is a risk factor of admission and mortality when a chemical is ingested intentionally. In the future, glacial acetic acid must be carefully managed, or its 100% concentrated formulation must be prohibited.

Additionally, ethylene glycol and methanol were identified as the risk factors for admission in both the unintentional and intentional ingestion groups in the present study. Ingestion of ethylene glycol and methanol has a serious prognosis because their metabolites, produced by alcohol dehydrogenase, have toxic effects, including metabolic acidosis, kidney injury (ethylene glycol), and visual loss (methanol). The 2017 AAPCC reported one case of methanol exposure resulting in death [[4]]. The administration of ethanol as an antidote should be performed for patients who ingest ethylene glycol or methanol [[15]].

This study has several limitations. First, detailed types of chemicals were not classified during the data collection phase. If more detailed information was obtained during the collection stage, the data would be more reliable. Second, insufficient information on the amounts and

concentrations of the chemicals is a limitation of this study. In the future, the amount of inges-tion should be included to the injury survey registry of the KCDC. Third, information on underlying diseases that could affect patient prognosis was insufficient. Fourth, the data collet-ing system, based on the ED, does not include all patients ingesting chemicals. Patients who did not receive hospital treatment or who used an outpatient clinic were excluded in the study.

Among patients aged older than 18 years who visited the ED due to chemical ingestion from 2011 to 2016, the most commonly ingested substances were hypochlorites, detergents, ethanol, and glacial acetic acid, and the type of substances ingested differed depending on pres-ence or absence of intentionality. High-concentration glacial acetic acid, a chemical substance that is commercially available in South Korea, is used in cooking, resulting in a significant number of glacial acetic acid ingestions, and is a risk factor for admission and mortality in the intentional ingestion group. The result of this study are considered beneficial in determining treatment policies for patients ingesting chemicals who presented to the ED and in managing and preventing chemical exposures.

## Supporting information

**S1 Table. Total incidence, admission and mortality of chemical ingestion in adult (>20 years) at ED, South Korea, 2011–2016.**
(DOCX)

**S2 Table. Univariate analysis of glacial acetic acid and age for mortality in intentional chemical ingestion.**
(DOCX)

## Acknowledgments

We are thanks to Hye Ah Lee. Ph.D. for helping statistical analysis.

## Author Contributions

**Conceptualization:** Duk Hee Lee.

**Formal analysis:** Jae Hee Lee.

**Funding acquisition:** Duk Hee Lee.

**Investigation:** Duk Hee Lee.

**Methodology:** Jae Hee Lee, Duk Hee Lee.

**Visualization:** Jae Hee Lee.

**Writing – original draft:** Jae Hee Lee, Duk Hee Lee.

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
