## [Decision Letter · Decision Letter 0]

13 Jan 2020

PONE-D-19-33190

Types and Clinical Outcomes of Chemical Ingestion in the Emergency Department: South Korea, 2011-2016

PLOS ONE

Dear Hee Lee, 

Thank you for submitting your manuscript to PLOS ONE. After careful consideration, we feel that it has merit but does not fully meet PLOS ONE’s publication criteria as it currently stands. Therefore, we invite you to submit a revised version of the manuscript that addresses the points raised during the review process. 

ACADEMIC EDITOR:  

This manuscript is not presented in a standard English, I suggest an English native speaker revision.

Statistical analysis was not well performed.

The main study aim were not clear.  I couldn't understand what’s the definition of intentional vs. unintentional groups.   

There are many comments from reviewers, and conflict between them. 

I ask you to produce a very well performed revision in order to publish this paper. 

Especially check the statistical corrections suggested, and try to reduce the tables, focusing on most important data or transferring them into supplementary materials. 

We would appreciate receiving your revised manuscript by February 3rd. To enhance the reproducibility of your results, we recommend that if applicable you deposit your laboratory protocols in protocols.io, where a protocol can be assigned its own identifier (DOI) such that it can be cited independently in the future. For instructions see: http://journals.plos.org/plosone/s/submission-guidelines#loc-laboratory-protocols

We look forward to receiving your revised manuscript.

Kind regards,

Martina Crivellari

Academic Editor

PLOS ONE

Journal Requirements:

2. We note that you have reported significance probabilities of 0 in places. Since p=0 is not strictly possible, please correct this to a more appropriate limit, eg 'p<0.0001'.

Reviewers' comments:

Reviewer's Responses to Questions

**Comments to the Author**

1. Is the manuscript technically sound, and do the data support the conclusions?

Reviewer #1: Yes

Reviewer #2: Yes

2. Has the statistical analysis been performed appropriately and rigorously? 

Reviewer #1: No

Reviewer #2: Yes

3. Have the authors made all data underlying the findings in their manuscript fully available?

Reviewer #1: Yes

Reviewer #2: Yes

4. Is the manuscript presented in an intelligible fashion and written in standard English?

Reviewer #1: Yes

Reviewer #2: No

5. Review Comments to the Author

Reviewer #1: The retrospective study reported the major types of chemical poisoning agents in patients who visited ED in South Korea from 2011 to 2016 and accessed the associations between hospital admission/mortality and potential risk factors including intentions of chemical ingestion, patients’ demographics characteristics, time from injury to ED visits, ED LOS, date, time, places of injury occurrence, and so on. My review/comments are from statistical perspective.

Study design:

Lines 74 – 75: the authors mentioned “all patients were classified into either intentional group or unintentional group.” What’s the definition of intentional vs. unintentional groups? Did patients in intentional group had psychiatric problems? Were those patients susceptible to suicide? The sample of intentional group maybe bias by only including patients who presented at ED.

The main study aim were not clearly mentioned. In the materials and methods section, it looks like the authors want to access the association between admitted/discharged and intentions. But, in the results section the authors reported results of parallel analyses on intentional group and unintentional group separately. I recommend the authors setup the primary aims and the corresponding hypotheses before analyzing the data.

Statistical Analysis:

The author didn’t clearly illustrate covariates used in multiple logistic regression. Did the authors perform any variable selection algorithms?

Period from time of injury to ED presentation may have crucial clinical meaning with respect to the primary endpoint (admitted/discharged). The measurement accuracy for intentional group maybe bias because of the potential bias in sampling.

In table 4, the author listed ingested chemicals with significant p-values in model. Did the author include single covariate (e.g. chemical agents) with all types of chemicals as different categories? If so, which chemical was defined as the reference level? Or did the author include all types of chemicals as dummy binary covariates? Is it possible that one patient ingested more than one chemicals?

In table 4, the authors reported “sulfuric acid” was statistically significant in the multivariable logistic regression and had a very large odds ratio and a huge upper limit (26.132 [3.105, 219.890]). It is confusing me that, in table 2, the admission rates of ingested “sulfuric acid” and “hydrofluoric acid” were 5.7% (n = 2) and 4.5% (n = 3) respectively. But “sulfuric acid” has such a huge odds ratio (OR=26.132) and “hydrofluoric acid” has odds ratio (OR=0.281) less than 1.

In addition, among 1673 unintentional ingestion cases only 35 (2.09%) ingested “sulfuric acid”. A quasi complete separation may occur and vague/suspected findings can be drawn.

Same issues in intentional group (both discharged vs. admitted and survival vs. mortality) as unintentional group.

I recommend authors pay more attention to interpret results of multivariable logistic regressions and move characteristics comparisons from result section to the discussion or supplementary section. For categorical characteristics with more than two categories, it is suspected and tedious to report p-values of Chi-square tests of each category without multiple adjustment. I recommend to perform one single Chi-square test for categorical characteristics and report a single p-value.

Additional comments:

Line 100 – 101: the authors reported three age groups with p-values < 0.05. Those p-values were separately calculated for each sub-group without showing literature evidence of associations between age groups and intentions of chemical ingestion. I recommend to exclude this sentence or do a multiple comparison adjustment in order to avoid erroneous inferences.

Table 1: how to explain different ED LOS?

Lines 121 – 122: the sentence is confused. The authors would like to report the top three unintentional chemical ingestions for admitted patients. In order to claim potential associations between admission and type of chemicals, admission rate would be better. Admission rates of ingesting hypochlorites (18.0%) and detergent (15.8%) were less than 20%. However, admission rates of ingesting ethylene glycol was 58.3%.

Lines 122 – 124: Why the authors ignored three death cases by ingesting toluene, methanol, and hydrochloric acid?

Lines 126 – 127: I recommend to report: the admission rate for male were statistically significantly higher than it for female. Report both admission rates.

Lines 127: Report both mean ages if it has clinical meanings.

Line 128: collapse insignificant levels and run a single Chi-square test to test association between age groups and admitted/discharged.

Table 4: it should be multivariable analysis.

Reviewer #2: General

Detailed and easy to understand article.

However, the main claims of the paper are of interest for South-Korea, mainly.

The data presented are less numerous than those already reported in the literature.

There are no data about the severity of outcome; the exposure concentrations / exposure doses.

There is no real novelty in comparison with previous literature.

The English style and language should be checked by an English native speaker.

Too many tables. A choice should be made to focus on the most important results. The remaining data could be available as Supplementary Material.

Details

Line 16: why were only ingestions and not inhalation or other routes of exposure also included?

Line 29: “OD” abbreviation: the first time the term is used, write it in full, please.

Line 32: “Differences in clinical outcomes by kinds of chemical, and intentionality” : these conclusions have already been reported in the literature

Line 33: “… study as a basis for treatment policies etc…”: policies already exist; e.g. Slaughter R’s publication in 2019 about the toxicology of sodium hypochlorite.

Lines 47-52: the aim of the article is relevant mainly for South-Korea.

Line 61: how many agencies are there in total in South-Korea? Are 20 or 23 agencies representative of the whole country? What is the served population in South-Korea?

Line 67: how many cases were excluded due to missing information on intentionality? And to missing on ED outcome?

Line 73: “categorized”: according to which classification?

Lines 76-77: “insurance type were compared etc…” why? What is the aim?

Line 79: “alcohol ingestion before injury…”: any data about the dose?

Line 89-90: “a logistic regression analysis was performed to determine the risks of hospital admission and mortality”: separate outcomes or composite outcome?

Line 90: “mortality”: in-hospital mortality?

Line 91: “p values less than 0.05”: use 95% Confidence Intervals instead of p values (more informative).

Line 94: (4741-2712)/4741 = 43%. Please, indicate the reason for exclusion of these 43% of cases.

Lines 109-111: “patients in the intentional group had significantly larger proportions of general ward admission, ICU care, and mortality /…/ than the unintentional ingestion group”: is this result really new ? Several articles have already reported that most fatalities after chemical ingestion in adults are intentional (e.g. CDC data from North America).

Table 1: “Age distribution in 10 years”, line “81”: add “≥” so as to get “≥ 81”

Table 1: “Mode of arrival”: what is the interest of this information? What is the aim?

Table 1: “Time interval from injury to ED visit (hrs)”: values between 12 and 13 hrs = very long ! and huge variation; what is the reason?

Table 1: “Insurance”: what is the interest of this information? Probably country dependent and difficult to generalize.

Table 1: “Place”: place of what? (where the poisoning occurred, I suppose?)

Table 4: “Age”: was age entered as a continuous variable?

Table 9: 4 rows i.e. “Non-Elderly, glacial acetic acid (-)” up to “Elderly, glacial acetic acid (+)”: consider including an interaction term “age * glacial acetic acid” in the regression model.

Line180-182: “The purpose of this study… affected the clinical results”: already said.

Lines 187-190: “the 2017 AAPCC annual report… in that order”: this report included all routes of exposures (not only ingestions); so, is it comparable?

6. PLOS authors have the option to publish the peer review history of their article (what does this mean?). If published, this will include your full peer review and any attached files.

Reviewer #1: No

Reviewer #2: No

---

## [Author Response · Author response to Decision Letter 0]

6 Feb 2020

ACADEMIC EDITOR: 

This manuscript is not presented in a standard English, I suggest an English native speaker revision.

Statistical analysis was not well performed.

The main study aim were not clear. I couldn't understand what’s the definition of intentional vs. unintentional groups. 

There are many comments from reviewers, and conflict between them. 

I ask you to produce a very well performed revision in order to publish this paper. 

Especially check the statistical corrections suggested, and try to reduce the tables, focusing on most important data or transferring them into supplementary materials. 

This manuscript underwent English editing before submission, but grammatical errors were still observed after editing. 

This revised manuscript edited by Editage English editing service. 

I attach the certification of editing. Thank you very much. 

The term “unintentional” in this study refers to an accident or assault by others, while “intentional” refers to a patient who attempted self-harm or suicide.

This study aimed to provide basic data on the types and frequency of chemical ingestions and the clinical outcomes of chemical ingestion injury. Based on several literature reports, intentional poisoning has a poor prognosis on admission and mortality. We compared the intentional and unintentional groups and analyze the factors affecting admission and mortality in each group. Each analysis is considered necessary considering the patients’ different characteristics. I will add these descriptions in the aim and hypothesis of this study.

Statistical analysis was reflected in the text and tables in consultation with the professor of statistics. 

I have simplified all the tables according based on your suggestion. Thank you for your advice.

Thank you for your kind review. We have thoroughly checked the manuscript and paid special consideration in addressing your suggestions. I appreciate you. Thank you very much. 

Journal Requirements:

The text has been revised to fit the template of PLOSOne.

2. We note that you have reported significance probabilities of 0 in places. Since p=0 is not strictly possible, please correct this to a more appropriate limit, eg 'p<0.0001'.

I modified P=0 to P<0.0001. Thank you very much. 

I linked my ORCID ID to my account. Thank you very much. 

Reviewers' comments:

Reviewer's Responses to Questions

Comments to the Author

1. Is the manuscript technically sound, and do the data support the conclusions?

Reviewer #1: Yes

Reviewer #2: Yes

2. Has the statistical analysis been performed appropriately and rigorously? 

Reviewer #1: No

Reviewer #2: Yes 

After reviewing the study with a statistician, I amended the statistics or statistical descriptions according to your advices. Thank you very much. 

 3. Have the authors made all data underlying the findings in their manuscript fully available?

Reviewer #1: Yes

Reviewer #2: Yes

4. Is the manuscript presented in an intelligible fashion and written in standard English?

Reviewer #1: Yes

Reviewer #2: No

This manuscript underwent English editing before submission, but grammatical errors were still observed after editing. 

 This Revised manuscript edited by Editage English editing service. 

 I attach the certification of editing. Thank you very much. 

5. Review Comments to the Author

Reviewer #1: The retrospective study reported the major types of chemical poisoning agents in patients who visited ED in South Korea from 2011 to 2016 and accessed the associations between hospital admission/mortality and potential risk factors including intentions of chemical ingestion, patients’ demographics characteristics, time from injury to ED visits, ED LOS, date, time, places of injury occurrence, and so on. My review/comments are from statistical perspective.

Thank you for your kind review. We have thoroughly checked the manuscript and have paid special consideration in addressing your suggestions.

Study design:

Lines 74 – 75: the authors mentioned “all patients were classified into either intentional group or unintentional group.” What’s the definition of intentional vs. unintentional groups? Did patients in intentional group had psychiatric problems? Were those patients susceptible to suicide? The sample of intentional group maybe bias by only including patients who presented at ED.

The term “unintentional” in this study refers to an accident or assault by others, while “intentional” refers to a patient who attempted self-harm or suicide.

 The Korea Centers for Disease Control and Prevention (KCDC) has been prospectively collecting injury-related information nationwide. The KCDC has been receiving real-time Emergency Department-Based Injury In-depth Surveillance information from 23 academic tertiary hospital emergency departments (EDs) nationwide. The KCDC has begun collecting the data of injury and poisoning in 2008 and has also been investigating chemical exposure cases. The KCDC data were examined for intentionality, but the causes of suicide attempts and psychiatric history were not investigated. South Korea had a suicide rate of 24.6 per 100,000 population, the highest among the Organisation for Economic Co-operation and Development (OECD) countries. The Korean Statistical Information Service dataset showed that suicide rates increased as people aged. I believe that it is necessary to investigate the patients’ psychiatric history of suicide/self-harm. The KCDC should investigate the patients’ psychiatric history and causes of self-harm because this information is considered potentially significant for public health.

Based on your suggestion, I also agree that there may be biases when only ED patients are included. However, a specialized center that treats poisoned patients in South Korea has not been established yet, and most of the poisoned patients are frequently encountered in the ED because the primary clinic does not treat these types of patients. Even in general hospitals, poisoned patients are referred to the ED. Most toxicologists in South Korea are emergency physicians. I am also an emergency physician, and I treat a poisoned patient. Nevertheless, depending on the severity of poisoning, there may be cases in which individuals are treated as outpatients. For patients who visit the ED, an association between the injury and chemical ingestion can be easily determined considering that patients are in the acute phase of the injury. Additionally, the severity of the patient in the ED is higher than that of an outpatient, so it is possible to investigate the injury that affects the patient’s poor prognosis. I will include the following as one of the limitations in this study: this study involved patients who visited the ED; therefore, these ED patients may have different characteristics to the generally poisoned patients.

The main study aim were not clearly mentioned. In the materials and methods section, it looks like the authors want to access the association between admitted/discharged and intentions. But, in the results section the authors reported results of parallel analyses on intentional group and unintentional group separately. I recommend the authors setup the primary aims and the corresponding hypotheses before analyzing the data.

This study aimed not only to compare the intentional and unintentional groups but also to analyze the factors affecting admission and mortality in each group. Based on several literature reports, intentional poisoning has a poor prognosis on admission and mortality. We analyze the factors that affect admission and mortality in each group and compare the intentional and unintentional ingestion groups. Each analysis is considered necessary considering the patients’ different characteristics. I will add these descriptions in the aim and hypothesis of this study.

Statistical Analysis:

The author didn’t clearly illustrate covariates used in multiple logistic regression. Did the authors perform any variable selection algorithms?

In this study, three multiple logistic regressions were performed. Using cross-analysis and average comparison, we selected the variables that showed significant differences between the two groups, and based on the researchers’ discretion, we selected the variables considered as factors for hospital admission and mortality. Multiple logistic regression using backward stepwise selection (likelihood ratio) was performed, and the remaining variables with p-value less than 0.05 were listed in the table.

Period from time of injury to ED presentation may have crucial clinical meaning with respect to the primary endpoint (admitted/discharged). The measurement accuracy for intentional group maybe bias because of the potential bias in sampling.

The intentionality of ingestion is determined by a well-trained emergency physician, and the data of this surveillance system are entered based on the physician’s medical records. The study is retrospective, and researchers are also medical staff who are not involved in the study. There is no investigator’s bias for each group.

We recheck the data. We found out that the average value is significantly high, and the standard deviation increases due to the presence of extreme values. The frequency distribution of the injury visit intervals is as follows. Since the data are obtained from 23 hospitals, the value should not be excluded. According to a statistician, it is more appropriate to present the median value due to the characteristics of the data. Therefore, the values in the table were modified based on the median values, and the Mann-Whitney U test was performed. The period from the time of injury to ED presentation was also entered when performing multivariate regression analysis for admission/discharge and mortality and was excluded as a significant factor affecting prognosis.

In table 4, the author listed ingested chemicals with significant p-values in model. Did the author include single covariate (e.g. chemical agents) with all types of chemicals as different categories? If so, which chemical was defined as the reference level? Or did the author include all types of chemicals as dummy binary covariates? Is it possible that one patient ingested more than one chemicals?

In this study, we reviewed the survey data of all chemical exposure cases registered in the Emergency Department-Based Injury In-depth Surveillance system operated by the KCDC. Information collected through the surveillance system contains the component or product name of the chemical that the patient ingested. Based on this, the ingredients with the largest proportion are classified as the main ingredients. This process was performed by one chemist and two emergency physicians. We used the dummy binary covariates to calculate the odds ratio (OR) for each chemical exposure.

In table 4, the authors reported “sulfuric acid” was statistically significant in the multivariable logistic regression and had a very large odds ratio and a huge upper limit (26.132 [3.105, 219.890]). It is confusing me that, in table 2, the admission rates of ingested “sulfuric acid” and “hydrofluoric acid” were 5.7% (n = 2) and 4.5% (n = 3) respectively. But “sulfuric acid” has such a huge odds ratio (OR=26.132) and “hydrofluoric acid” has odds ratio (OR=0.281) less than 1.

It is not sulfuric acid but hydrogen sulfide. My sincere apology as there was a mistake in the English translation, and it was not comprehensively reviewed. Thank you for your careful review. A total of seven cases of hydrogen sulfide ingestion in the unintentional group were reported, of which six were hospitalized, showing an 85.7% admission rate. This can be checked in Supplement 1.

In addition, among 1673 unintentional ingestion cases only 35 (2.09%) ingested “sulfuric acid”. A quasi complete separation may occur and vague/suspected findings can be drawn.

Same issues in intentional group (both discharged vs. admitted and survival vs. mortality) as unintentional group.

Thank you for pointing out the typographical error regarding hydrogen sulfate as previously stated. We wanted to present significantly comprehensive details regarding the type of chemicals ingested and to provide accurate data for patients’ clinical care. In intentional poisoning, sulfuric acid is not commonly observed. It is believed that sulfuric acid is a substance that cannot be easily accessed by common individuals. However, sulfuric acid can be possibly purchased in Korea. Moreover, glacial acetic acid, which is a significantly strong acid, can also be found in convenience stores in Korea. Most unintentional sulfuric acid poisoning is observed in factories.

I recommend authors pay more attention to interpret results of multivariable logistic regressions and move characteristics comparisons from result section to the discussion or supplementary section. For categorical characteristics with more than two categories, it is suspected and tedious to report p-values of Chi-square tests of each category without multiple adjustment. I recommend to perform one single Chi-square test for categorical characteristics and report a single p-value.

For variables that do not have significant clinical significance, we presented a single p-value.

Additional comments:

Line 100 – 101: the authors reported three age groups with p-values < 0.05. Those p-values were separately calculated for each sub-group without showing literature evidence of associations between age groups and intentions of chemical ingestion. I recommend to exclude this sentence or do a multiple comparison adjustment in order to avoid erroneous inferences.

A multiple comparison adjustment was not performed in this study. Hence, the sentence was already deleted. Thank you for your suggestion.

Table 1: how to explain different ED LOS?

According to Table 1, the proportions of general ward admission, intensive care unit care, and death are significantly high in the intentional group. As shown in Tables 3, 6, and 8, the ED length of stay (LOS) is significantly longer in the admission group than in the discharge group, and the ED LOS is significantly longer in the mortality group than in the survivor group. Significantly high rates of admission and death in the intentional group are possibly attributed to longer ED LOS.

Lines 121 – 122: the sentence is confused. The authors would like to report the top three unintentional chemical ingestions for admitted patients. In order to claim potential associations between admission and type of chemicals, admission rate would be better. Admission rates of ingesting hypochlorites (18.0%) and detergent (15.8%) were less than 20%. However, admission rates of ingesting ethylene glycol was 58.3%.

I agree with your comment. I modified the manuscript showing the high ingestion incidence and high admission rates. The high incidence of admission is associated with the incidence of ingestion, and the high admission rate is possibly associated with chemical toxicity. The results were presented because identifying the frequency of ingestion and managing the chemical and reducing the incidence of chemical ingestion in patients are also significant.

I added the following sentence: “The admission rates were as follows: 60% for glacial acetic acid, 58.3% ethylene glycol, and 30.4% other alkali agents.”

Lines 122 – 124: Why the authors ignored three death cases by ingesting toluene, methanol, and hydrochloric acid?

We did not describe the information presented in the table because we believed that repeatedly describing this information in the main text would result in redundancy. Nevertheless, we will add a description regarding these substances to avoid misunderstanding. Hence, toluene (1), methanol (1), and hydrochloric acid (1) were added. Thank you for your suggestion.

Lines 126 – 127: I recommend to report: the admission rate for male were statistically significantly higher than it for female. Report both admission rates.

I added the following sentence in the main text: “23.04% of men and 17.74% of women were admitted.”

Lines 127: Report both mean ages if it has clinical meanings.

I added the following in the main text: “The mean age of the admission group was higher (54.01 years) than that of the discharge group (45.59 years).”

Line 128: collapse insignificant levels and run a single Chi-square test to test association between age groups and admitted/discharged.

We believed that presenting the distribution by age group would help in identifying the frequency of poisoning and in establishing preventive measures. A single chi-squared test was performed based on your suggestion.

Table 4: it should be multivariable analysis.

Multiple logistic regression using backward stepwise selection (likelihood ratio) was performed by selecting variables that showed a significant difference between the two groups using cross-analysis and average comparison. Subsequently, the remaining variables with p-value less than 0.05 were listed in the table. We added the result of the univariate analysis in Table 4.

Reviewer #2: 

. 

I would like to thank you for your careful review and suggestions on my paper. I have closely reviewed and answered the points you pointed out.

General

Detailed and easy to understand article. 

However, the main claims of the paper are of interest for South-Korea, mainly. 

The KCDC has been prospectively collecting injury-related information nationwide. The KCDC has been receiving real-time Emergency Department-Based Injury In-depth Surveillance information from 23 academic tertiary hospital EDs nationwide. The KCDC has begun collecting the data of injury and poisoning in 2008 and has also been investigating chemical exposure cases. There are 36 regional emergency medical centers and 116 local emergency medical centers in Korea. A total of 36 regional emergency medical centers can treat poisoned patients. The KCDC has provided financial support to 23 regional emergency medical centers and maintained a high-quality patient information by sending one coordinator to each hospital. My hospital is one of the 23 hospitals, and we requested and received high-quality information from the KCDC. We studied and analyzed this information retrospectively. Hence, this study aimed to determine the types and clinical outcomes of chemical ingestion in South Korea. 

The types and frequencies of chemical exposures may vary from country to country, which is dependent on the country’s development. South Korea’s nationwide information regarding chemical ingestion is considered significant.

I believe it would be more significant, but practically difficult, if some developing countries conduct research together.

Developed countries’ emergency centers, including the Poison Control Center (PCC) in the United States and the Canadian Transport Emergency Center in Canada, operate an addiction management system.

According to the American Chemical Society, approximately 246,000 chemicals are commercially distributed worldwide, with 40,731 chemicals being distributed in Korea and approximately 400 new chemicals entering the Korean domestic market annually.

As the use of chemicals increases, exposure to chemicals or chemical products also increases. Moreover, the types of chemicals are evolving. Accordingly, it is necessary to establish an information system for public and healthcare providers. However, Korea does not have an emergency poison response or control center.

The KCDC collects data regarding the chemical types and exposure routes. Ingestion was the most frequent exposure route, with intentional exposure being the most common.

The authors work in the emergency medical center, teach medical students, and admit toxically poisoned patients. In Korea, these data are not published. Recently, we believe that a study should be conducted on chemical ingestion considering the insufficient information on the frequencies, types, and clinical outcomes of chemical ingestion.

One of the limitations in this study was as follows: a specific amount of chemical ingestion was not assessed in this study.

Hence, it is believed that the patients should be classified by intentionality, the unintentionally poisoned and intentionally poisoned patients.

It is already known that intentional poisoning is more fatal than unintentional poisoning. Korea has the highest suicide rate among the OECD countries, and the number of suicide attempts was 28,278 in 2017, according to the Korea Suicide Prevention Center. It was significant to divide the patients into the unintentional and intentional ingestion groups to assess their clinical results.

Data regarding the types, frequencies, and clinical outcomes of chemical ingestion are considered beneficial in establishing health policies, determining the patient’s prognosis, and educating patients regarding chemical ingestion.

The data presented are less numerous than those already reported in the literature.

The data analyzed a total of 4,741 chemical ingestions based on the 1,537,617 injured and addicted patients from 2011 to 2016. Although the amount of chemical ingestion may be less than that of the United States or Europe, the data from the 23 nationwide emergency medical centers in a country comprising 50 million people are considered significant.

There are no data about the severity of outcome; the exposure concentrations / exposure doses. There is no real novelty in comparison with previous literature.

Mortality and length of hospital stay are considered the risk factors for chemical ingestion. Chemical exposure in everyday life is increasing, and the types of chemicals used in industrial products are increasing in developing countries such as Korea. The results regarding chemical ingestion in the past 5 years are considered significant. As previously stated, the absence of a specific amount of chemical ingestion was considered a limitation in this study.

The English style and language should be checked by an English native speaker. 

This manuscript underwent English editing before submission, but grammatical errors were still observed after editing. This revised document was edited by Editage Editing service. (I attach the certification of editing)

Too many tables. A choice should be made to focus on the most important results. The remaining data could be available as Supplementary Material.

I have simplified all the tables according based on your suggestion. Thank you for your advice.

Details

Line 16: why were only ingestions and not inhalation or other routes of exposure also included?

Oral ingestion is the most frequent routes of poisoning in the clinical field. In chemical poisoning, ingestion is considered to have a more serious prognosis than other routes of exposure. Ingestion of various chemicals is quickly absorbed by the body, resulting in the uncertainty of prognosis. According to KCDC’s input classification, chemical exposure is divided into solid, liquid, gas, and other unknown classifications. Moreover, exposure routes are divided into ingestion and other exposure routes.

 Gas accounted for 20% of all chemical poisonings. Regarding gaseous chemicals, most of the exposure route was inhalation. A total of 90.4% of gas exposure were from carbon monoxide (CO). If gas was reported, it is likely to be in the form of a single substance. Hence, a study assessing chemical ingestion through the oral route was conducted. In addition, CO poisoning has a well-established literature, treatment, and prognosis prediction.

Line 29: “OD” abbreviation: the first time the term is used, write it in full, please.

I modified the full term and abbreviation. I changed OD to OR.

Line 32: “Differences in clinical outcomes by kinds of chemical, and intentionality” : these conclusions have already been reported in the literature

We are aware that there are several papers and case reports regarding chemical exposure. Our data are considered beneficial in determining the types of chemicals that are currently poisoning the industrialized countries with a population of 50 million and in assessing the mortality rate. Certainly, according to the reviewer, assessing the poisonous doses and chemical concentrations in the blood is considered significant.

Line 33: “… study as a basis for treatment policies etc…”: policies already exist; e.g. Slaughter R’s publication in 2019 about the toxicology of sodium hypochlorite.

There are several existing studies and case reports regarding chemical ingestion.

Slaughter R’s publication has conducted a review and case reports regarding sodium hypochlorite for 68 years.

This study also assesses the frequency and mortality of the frequently ingested chemicals.

We believe that this study can provide sufficient information on the recent toxic substances and patients’ clinical prognosis.

Lines 47-52: the aim of the article is relevant mainly for South-Korea.

Korea is a developing country that has progressed significantly over the past 50 years, recently ranking as top 10 in terms of gross domestic product. The type and frequency of chemical exposures may vary from country to country, which is dependent on the country’s development. South Korea’s nationwide information regarding chemical ingestion is considered significant.

I think it would be more significant, but practically difficult, if some developing countries conduct research together.

Line 61: how many agencies are there in total in South-Korea? Are 20 or 23 agencies representative of the whole country? What is the served population in South-Korea?

Korea has a total of 51,780,000 individuals living in a small land area.

In Korea, there are 36 regional emergency medical centers (Level 1), 116 local emergency medical centers (Level 2), and approximately 150 local emergency medical rooms. Regional emergency medical centers and some local emergency medical centers can treat poisoned patients.

The KCDC has been receiving real-time Emergency Department-Based Injury In-depth Surveillance information from the 23 academic tertiary hospital EDs (regional emergency medical centers) nationwide. It is not possible to represent all injuries in Korea, but Korea has detailed information regarding injuries. Hence, an in-depth analysis of injured patients is considered significant.

Line 67: how many cases were excluded due to missing information on intentionality? And to missing on ED outcome?

Children (1247) and individuals with unknown ED results (644) and unknown intentionality (138) were excluded according to the exclusion criteria.

Line 73: “categorized”: according to which classification?

We used the classification of the 2017 Annual Report of the American Association of PCCs’ National Poison Data System: 35th Annual Report.

We reviewed the data from all chemical exposure cases of the KCDC. Two emergency physicians and one chemist identified and categorized the main components of the ingested chemicals.

Lines 76-77: “insurance type were compared etc…” why? What is the aim?

South Korea had a suicide rate of 24.6 per 100,000 population, the highest among the OECD countries.

South Korea is a country where national health insurance is available to all individuals. Among them, those who are experiencing financial difficulties receive Medicaid care (the country pays all the healthcare costs).

The analysis was performed because the type of insurance, that is, the economic situation of the patient, is possibly associated with intentional chemical ingestion. However, a statistically significant difference regarding the type of insurance was not observed.

Line 79: “alcohol ingestion before injury…”: any data about the dose?

Whether the patient has consumed alcohol or not is investigated. Data regarding the amount of alcohol are not provided in this study.

My hospital is one of the 23 hospitals that send information to the KCDC. For patients in the hospital, they do not undergo screening for alcohol use using a complete Alcohol Use Disorders Identification Test, or their blood alcohol concentration (BAC) is not assessed. The KCDC has no available information regarding BAC. Although ideally the BAC levels should be measured, difficulties with privacy, cost, and ethical issues usually arise when measuring the BAC. I instructed the KCDC to measure the patients’ self-reported amount of alcohol (in addition to the BAC level). According to the KCDC, government funding is required to accurately measure the patients’ alcohol consumption.

Line 89-90: “a logistic regression analysis was performed to determine the risks of hospital admission and mortality”: separate outcomes or composite outcome?

These were considered as separate outcomes for each hospital admission and mortality.

Line 90: “mortality”: in-hospital mortality?

It was in-hospital mortality.

The table shows that mortality in the ED is defined as mortality in the emergency room during treatment after arriving at the ED. Hospital mortality is defined as any death in the hospital, including ED death.

An explanation regarding this will be provided below.

Line 91: “p values less than 0.05”: use 95% Confidence Intervals instead of p values (more informative).

I modified this sentence in the Statistics section based on your suggestion. Thank you for your advice.

Line 94: (4741-2712)/4741 = 43%. Please, indicate the reason for exclusion of these 43% of cases.

Children (1247) and individuals with unknown ED results (644) and unknown intentionality (138) were excluded according to the exclusion criteria.

Lines 109-111: “patients in the intentional group had significantly larger proportions of general ward admission, ICU care, and mortality /…/ than the unintentional ingestion group”: is this result really new? Several articles have already reported that most fatalities after chemical ingestion in adults are intentional (e.g. CDC data from North America).

Intentional ingestion has a high rate of admission and mortality, even if the patient is addicted to the same substance. It is considered that patients who intentionally ingest chemicals ingest a large amount of chemicals. Although measuring blood levels would be accurate, each chemical does not have an accurate test. According to other reports, intentional poisoning has a high mortality rate without measuring the patients’ blood levels.

As reported in several literatures, intentional addiction has a poor prognosis in admission and mortality. The results of our study were expectedly significant in the intentional ingestion group. We analyze the factors that affect admission and mortality in each group and compare the intentional/unintentional ingestion groups. Each analysis is considered necessary considering the patients’ different characteristics.

I believe that the classifications and clinical outcomes of chemical ingestion in this study are more specific than in other studies.

Table 1: “Age distribution in 10 years”, line “81”: add “≥” so as to get “≥ 81”

I added “≥.” Thank you for your suggestion.

Table 1: “Mode of arrival”: what is the interest of this information? What is the aim?

“Mode of arrival” is a variable of the KCDC. 119 (which is equivalent to 911 in North America) is free of charge. The use of 119 is the public cost of the country. The intentional ingestion group was expected to use more 119 due to severe clinical symptoms than the unintentional ingestion group. However, there was no statistical difference between the two groups.

Table 1: “Time interval from injury to ED visit (hrs)”: values between 12 and 13 hrs = very long ! and huge variation; what is the reason?

We rechecked the data. We found out that the average value is significantly high, and the standard deviation increases due to the presence of extreme values. The frequency distribution of the injury visit intervals was as follows. Since the data are obtained from 23 hospitals, the value should not be excluded. According to a statistician, it is more appropriate to present the median value due to the characteristics of the data. Therefore, the values in the table were corrected based on the median values, and Mann-Whitney U test was performed. The period from the time of injury to ED presentation was also entered when performing multivariate regression analysis for admission/discharge and mortality and were excluded as a significant factor affecting prognosis. I modified the table.

Table 1: “Insurance”: what is the interest of this information? Probably country dependent and difficult to generalize.

I agree that it is difficult to generalize the insurance types because insurance systems vary from country to country.

South Korea is a country where national health insurance is available to all individuals. Among them, those who are experiencing financial difficulties receive Medicaid care (the country pays all the healthcare costs).

The analysis was performed because the type of insurance, that is, the economic situation of the patient, is possibly associated with intentional chemical ingestion. However, a statistically significant difference regarding the type of insurance was not observed.

Table 1: “Place”: place of what? (where the poisoning occurred, I suppose?)

I agree with your suggestion. Hence, the phrase was changed to “where the poisoning occurred.”

Table 4: “Age”: was age entered as a continuous variable?

Age was entered as a continuous variable.

Table 9: 4 rows i.e. “Non-Elderly, glacial acetic acid (-)” up to “Elderly, glacial acetic acid (+)”: consider including an interaction term “age * glacial acetic acid” in the regression model.

I performed an analysis using age * glacial acetic acid and mentioned the p-value in the text. Moreover, the table containing “Elderly, glacial acetic acid” was moved to the supplementary table. Thank you for your suggestion.

Line180-182: “The purpose of this study… affected the clinical results”: already said.

This phrase was already mentioned in the previous text. Hence, I will delete this per your suggestion.

Lines 187-190: “the 2017 AAPCC annual report… in that order”: this report included all routes of exposures (not only ingestions); so, is it comparable?

 � The AAPCC reports all poisoning substances and all routes. It is not only about chemical ingestion. However, the poisoning of all substances was considered comparable with the most extensively reported systematic report. This study presented more information regarding the types and mortality rates of chemical ingestion than our study.

---

## [Decision Letter · Decision Letter 1]

19 Feb 2020

Types and clinical outcomes of chemical ingestion in emergency departments in South Korea (2011-2016)

PONE-D-19-33190R1

Dear Dr. Hee Lee,

We are pleased to inform you that your manuscript has been judged scientifically suitable for publication and will be formally accepted for publication once it complies with all outstanding technical requirements.

With kind regards,

Martina Crivellari

Academic Editor

PLOS ONE

Additional Editor Comments (optional):

Reviewers' comments:

Reviewer's Responses to Questions

**Comments to the Author**

1. If the authors have adequately addressed your comments raised in a previous round of review and you feel that this manuscript is now acceptable for publication, you may indicate that here to bypass the “Comments to the Author” section, enter your conflict of interest statement in the “Confidential to Editor” section, and submit your "Accept" recommendation.

Reviewer #1: All comments have been addressed

2. Is the manuscript technically sound, and do the data support the conclusions?

Reviewer #1: Yes

3. Has the statistical analysis been performed appropriately and rigorously? 

Reviewer #1: Yes

4. Have the authors made all data underlying the findings in their manuscript fully available?

Reviewer #1: Yes

5. Is the manuscript presented in an intelligible fashion and written in standard English?

Reviewer #1: Yes

6. Review Comments to the Author

Reviewer #1: The authors quickly responded all my questions/comments, adjust statistical methods, modified the conclusion, and further discussed limitations. A lot of work have been done. I have no further comments.

7. PLOS authors have the option to publish the peer review history of their article (what does this mean?). If published, this will include your full peer review and any attached files.

Reviewer #1: No

---

## [Editor Report · Acceptance letter]

26 Feb 2020

PONE-D-19-33190R1 

Types and clinical outcomes of chemical ingestion in emergency departments in South Korea (2011-2016) 

Dear Dr. Lee:

I am pleased to inform you that your manuscript has been deemed suitable for publication in PLOS ONE. Congratulations! Your manuscript is now with our production department. 

With kind regards,

on behalf of

Dr. Martina Crivellari 

Academic Editor

PLOS ONE